# Development and validation of an end stage kidney disease awareness survey: Item difficulty and discrimination indices

Tatiana Orozco[1], Emma Segal[2☯], Colin Hinkamp[2☯], Olanrewaju Olaoye[2], Popy Shell[1], Ashutosh M. Shukla[1,2]*

1 North Florida / South Georgia Veteran Healthcare System, Gainesville, FL, United States of America,
2 Division of Nephrology, Hypertension and Transplantation, University of Florida, Gainesville, FL, United States of America

☯ These authors contributed equally to this work.
* Ashutosh.Shukla@medicine.ufl.edu

## Abstract

### Introduction

Lack of awareness for chronic kidney disease (CKD), including end stage kidney disease (ESKD) and their management options is a major impediment to patients being able to select and use home dialysis therapies. While some instruments have been developed to measure CKD awareness, we lack validated instruments to evaluate patients' awareness of ESKD and dialysis modalities. This study is part of multipart project for developing and validating an ESKD-centric disease awareness questionnaire.

### Methods

A team of specialty renal care experts developed a 45-items questionnaire encompassing the subdomains of General Kidney Knowledge, CKD Knowledge, and ESKD Knowledge. Item reduction analysis—specifically, calculation of item difficulty and item discrimination index scores—was used to items for further review and potential removal.

### Results

Index scores were reviewed in conjunction with consideration of theoretical and substantive item content to reduce the number of items in the questionnaire, resulting in a 32-item questionnaire, retaining 5/10 items in the general kidney knowledge subdomain, 14/21 items in the CKD knowledge subdomain, and 13/14 items in the ESKD knowledge subdomain. Retained items ranged from 0.19 to 0.79 on the difficulty index, and from 0.31 to 0.81 on the discrimination index. Scores for percent correct on the reduced questionnaire spanned 0% to 87.5% correct on the full scale, 0% to 100% correct on the General Knowledge subdomain, 0% to 100% on the CKD Knowledge subdomain, and 0% to 92.3% on ESKD Knowledge subdomain.

**Data Availability Statement:** All relevant data are within the paper and its Supporting Information files.

**Funding:** Shukla AM reports funding from Department of Veterans Affairs, Health Service Research and Development (I01HX002639), Office of Rural Health (16004), and Clinical Science Research and Development (I01CX001661). The funders had no role in study design, data collection and analysis, decision to publish, or preparation of the manuscript.

**Competing interests:** The authors have declared that no competing interests exist.

## Conclusions

The questionnaire developed and refined in this study constitutes a patient disease awareness instrument that spans a range of difficulty, and yet shows strong ability to distinguish between patients with varying levels of disease awareness. This study is the first in part of a multistep project to addresses a gap in measuring ESKD specific knowledge. Accurate assessment of patients' disease awareness through a validated instrument can allow identification of knowledge domains leading to positive impacts on their healthcare decisions and improve targeted patient education efforts.

## Introduction

Over 97% of the US incident End-Stage Renal Disease (ESKD) patients are managed by some form of dialysis therapy [1]. Considering that home modalities of dialysis provide equivalent survival and trends for better patient-centered outcomes at a significantly lower healthcare cost (relative to in-center dialysis) [2–7], overwhelming use of in-center dialysis for the management of ESKD in the US has been a concern for over two decades. Thus, most stakeholders in ESKD, including the Center for Medicare and Medicaid Services have repeatedly advocated for increasing informed home dialysis utilization for the management ESKD [8, 9].

However, an individual's dialysis modality is conventionally recommended to be finalized through patient empowerment and shared decision-making. Informed dialysis selection requires individual patients to comprehend the complex medical, social, and financial aspects of their dialysis options and select the modality of dialysis best suited to their life-style [10–13]. Prior studies have shown that providing comprehensive pre-ESKD education (CPE) to patients with advanced chronic kidney disease (CKD) results in significant increase in patient-centered selection and utilization of home dialysis therapies [14–16]. Longitudinal cohort studies have further shown that such CPE further improves multiple pre-ESKD and post-ESKD outcomes including survivals [17–19]. Unfortunately, providing CPE and affecting patient-centered transition to ESKD requires substantial resources that are poorly compensated by the current reimbursement standards. As a result, few nephrology providers are able to provide CPE to all of their advanced CKD patients [20].

Several investigators have developed survey instruments to assess CKD awareness and knowledge with high degree of reliability and validity [21]. Unfortunately, most of these CKD awareness instruments primarily focus on CKD and have limited items to evaluate the comprehension and awareness of ESKD, including various dialysis modalities. Determining the domains of kidney disease knowledge that are predictive of facilitating patients' ability to reach informed dialysis decision, and specifically home dialysis selection, may allow better targeting of the educational efforts in advanced CKD patients. Assessment of ESKD knowledge specifically is also important because it will allow examination of which facets of patient ESKD knowledge are associated with increased informed dialysis selection and increased home dialysis use.

The overarching purpose of this study is to conduct a multistep validation process for a questionnaire focused on knowledge and awareness of ESKD, then further evaluate if awareness predicts outcomes in patients' selection of dialysis modality, and which disease awareness domains are associated with those outcomes. The current study focuses on item reduction analysis—specifically, item difficulty and item discrimination—to evaluate the utility of an initial set of items developed for our questionnaire assessing patients' kidney disease knowledge.

## Methods

The data obtained for this study are a part of a prospective observatory cohort study aimed at evaluating the effects of CKD patients' knowledge and awareness about kidney disease on their HRQoL and clinical outcomes. The study was approved by the Institutional Review Board at University of Florida. In the first phase of the study, 108 consecutive consenting adults, aged 18 years or older, attending the university specialty nephrology clinic were enrolled after obtaining written informed consent. Baseline assessments included questionnaire for self-reported socioeconomic factors, Charlson Comorbidity Index, HRQoL as measured by the Kidney Disease Quality of Life (KDQoL-36) without the dialysis item, the study questionnaire for kidney disease knowledge assessment, and their renal replacement therapy choices if there was an immediate hypothetical need for initiating dialysis therapy. Once the baseline data was collected, participants are followed prospectively at bi-annual intervals through the electronic health record (EHR) surveillance for major cardiovascular and renal outcomes.

To measure kidney disease knowledge, a team of experts from all parts of specialty renal care team created items with a prompt of "what are the critical bits of information that an advanced CKD patient should know?" This team included a renal dietician, renal social worker, pharmacist, dialysis nurse, and renal providers, which included nephrologist and an advanced nurse practitioner specializing in the care of advanced CKD and ESKD. Multiple-choice questions were designed to span a broad range of difficulty. For example, "which one of the following [response options] is one of the most common causes of kidney failure world-wide?" was designed as a relatively easy question, whereas "if you get Medicare and decide to do in-center hemodialysis, when does your Medicare begin?" was designed as a relatively difficult question.

In total, the authors developed a 45-item questionnaire whose items spanned 9 lower-level topics and could be grouped across three broader, higher-level subdomains: general kidney knowledge, CKD knowledge, and ESKD knowledge (see Table 1). All items were multiple-choice, with four to five total response options and one correct option for each item. While the items were multiple choice, they were scored dichotomously as either correct or incorrect, with item score = 1 for a correct response and item score = 0 for an incorrect response. Missing items scores—items where a participant did not answer—were scored as incorrect. Item scores were then summed within each subdomain to form subdomain scores. Item scores for the entire 45-item scale were summed to form a total kidney disease awareness score.

**Table 1. Number of items within topics, within scale subdomains.**

| Item Topics | Items within Subdomains | | | |
| --- | --- | --- | --- | --- |
| | General | CKD | ESKD | Total* |
| Terminology | 6 | 2 | 5 | 13 |
| Function | 3 | 9 | 0 | 12 |
| Laboratory test | 1 | 8 | 0 | 9 |
| Etiology | 2 | 0 | 0 | 2 |
| Access | 0 | 0 | 4 | 4 |
| Treatment | 2 | 8 | 11 | 21 |
| Lifestyle | 1 | 3 | 7 | 11 |
| Correlative Understandings | 2 | 8 | 4 | 14 |
| Healthcare Finances | 0 | 0 | 2 | 2 |

*Items within topics may be relevant to more than one subdomain. Thus, the aggregate number of the focus areas is greater than the total items.

Item reduction analysis was used to refine and reduce the number of items in a kidney disease knowledge questionnaire. The primary item reduction indices used to identify candidate deletion items were item difficulty and item discrimination. These indices were calculated for the total kidney disease awareness score and for the three subdomain scores.

Item difficulty, perhaps better understood as item easiness, is the proportion of correct responses on the item across the total sample. It is calculated as $c/n$, where $c$ is the number of correct responses for the item, and $n$ is the number of participants (blank responses were scored as incorrect; Fletcher, 2010). Items were categorized into five groups based on item difficulty (or easiness) score, ranging from "too difficult" (0 to .20) to "too easy" (.80 to 1.0) (Table 2). Given that this is a general-purpose scale for assessing a patient's awareness of their kidney disease state—as opposed to, for example, a scale which is designed to detect expertise, or otherwise be highly sensitive at the lower and upper ends of disease knowledge—thresholds of .20 and .80 were used for item difficulty. In other words, items where less than 20% or more than 80% of the respondents scored correctly, regardless of their total or subdomain kidney disease awareness scores, were identified for further review and potential removal.

Item discrimination is the item's ability to distinguish between respondents who score high vs respondents who score low on a subdomain or on the total score. Item discrimination is

**Table 2. Item difficulty (Easiness).**

| | Original Scale | | Reduced Scale | |
|---|---|---|---|---|
| | *n (%)* | *M (SD)* | *n (%)* | *M (SD)* |
| Full Scale | | | | |
| Item Difficulty Index (Easiness) | 45 | .44 (.23) | 32 | .47 (.45) |
| ≤ .20 (Too difficult) | 9 (20%) | – | 1 (3.1%) | – |
| 21 - .40 (Difficult) | 9 (20%) | – | 8 (25.0%) | – |
| 41 - .60 (Moderate) | 16 (35.6%) | – | 16 (50.0%) | – |
| 61 - .79 (Easy) | 7 (15.6%) | – | 7 (2.2%) | – |
| > .80 (Too easy) | 4 (8.9%) | – | 0 (0.0%) | – |
| Subdomain: General Kidney Knowledge | | | | |
| Item Difficulty Index (Easiness) | 10 | .68 (.22) | 5 | .64 (.12) |
| ≤ .20 (Too difficult) | 1 (10.0%) | – | 0 (0.0%) | – |
| 21 - .40 (Difficult) | 0 (0.0%) | – | 0 (0.0%) | – |
| 41 - .60 (Moderate) | 2 (20.0%) | – | 2 (40.0%) | – |
| 61 - .79 (Easy) | 3 (30.0%) | – | 3 (60.0%) | – |
| > .80 (Too easy) | 4 (40.0%) | – | 0 (0.0%) | – |
| Subdomain: CKD Knowledge | | | | |
| Item Difficulty Index (Easiness) | 21 | .38 (.22) | 14 | .51 (.14) |
| ≤ .20 (Too difficult) | 6 (28.6%) | – | 0 (0.0%) | – |
| .21 - .40 (Difficult) | 3 (14.3%) | – | 2 (14.3%) | – |
| .41 - .60 (Moderate) | 8 (38.1%) | – | 8 (57.1%) | – |
| .61 - .79 (Easy) | 4 (19.0%) | – | 4 (28.6%) | – |
| > .80 (Too easy) | 0 (0.0%) | – | 0 (0.0%) | – |
| Subdomain: ESKD Knowledge | | | | |
| Item Difficulty Index (Easiness) | 14 | .35 (.13) | 13 | .38 (.10) |
| ≤ .20 (Too difficult) | 2 (14.3%) | – | 1 (7.7%) | – |
| .21 - .40 (Difficult) | 6 (42.9%) | – | 6 (46.2%) | – |
| .41 - .60 (Moderate) | 6 (42.9%) | – | 6 (46.2%) | – |
| .61 - .79 (Easy) | 0 (0.0%) | – | 0 (0.0%) | – |
| > .80 (Too easy) | 0 (0.0%) | – | 0 (0.0%) | – |

**Table 3. Item discrimination index summaries.**

| | Original Scale | | Reduced Scale | |
|---|---|---|---|---|
| | *n (%)* | *M (SD)* | *n (%)* | *M (SD)* |
| **Full Scale** | | | | |
| Item Discrimination | 45 | .45 (.21) | 32 | .55 (.14) |
| ≤ 0 (Negatively & non-discriminating) | 0 (0%) | – | 0 (0.0%) | – |
| .01 - .20 (Poor) | 7 (15.6%) | – | 0 (0.0%) | – |
| .21-.29 (Marginal, Acceptable) | 1 (2.2%) | – | 1 (3.1%) | – |
| .30 - .39 (Good) | 8 (17.8%) | – | 3 (9.4%) | – |
| ≥ .40 (Excelllent) | 29 (64.4%) | – | 28 (87.5%) | – |
| **Subdomain: General Kidney Knowledge** | | | | |
| Item Discrimination | 10 | .50 (.13) | 5 | .59 (.10) |
| ≤ 0 (Negatively & non-discriminating) | 0 (0.0%) | – | 0 (0.0%) | – |
| .01 - .20 (Poor) | 0 (0.0%) | – | 0 (0.0%) | – |
| .21-.29 (Marginal, Acceptable) | 1 (10.0%) | – | 0 (0.0%) | – |
| .30 - .39 (Good) | 1 (10.0%) | – | 0 (0.0%) | – |
| ≥ .40 (Excelllent) | 8 (80.0%) | – | 5 (100.0%) | – |
| **Subdomain: CKD Knowledge** | | | | |
| Item Discrimination | 21 | .46 (.23) | 14 | .62 (.11) |
| ≤ 0 (Negatively & non-discriminating) | 0 (0.0%) | – | 0 (0.0%) | – |
| .01 - .20 (Poor) | 4 (19.0%) | – | 0 (0.0%) | – |
| .21-.29 (Marginal, Acceptable) | 1 (4.8%) | – | 0 (0.0%) | – |
| .30 - .39 (Good) | 3 (14.3%) | – | 0 (0.0%) | – |
| ≥ .40 (Excelllent) | 13 (61.9%) | – | 14 (100.0%) | – |
| **Subdomain: ESKD Knowledge** | | | | |
| Item Discrimination | 14 | .59 (.20) | 13 | .63 (.15) |
| ≤ 0 (Negatively & non-discriminating) | 0 (0.0%) | – | 0 (0.0%) | – |
| .01 - .20 (Poor) | 1 (7.1%) | – | 0 (0.0%) | – |
| .21-.29 (Marginal, Acceptable) | 0 (0.0%) | – | 0 (0.0%) | – |
| .30 - .39 (Good) | 0 (0.0%) | – | 0 (0.0%) | – |
| ≥ .40 (Excelllent) | 13 (92.9%) | – | 13 (100.0%) | – |

calculated as *(u-l)/n*, where *u* is the score correct by respondents who rate in the upper 1/3 of correct scores, *l* is the score correct by respondents who rate in the lowest 1/3 of the correct scores, and *n* is the total number of respondents [22]. Following guidelines and examples from Crocker & Algina (1986) and Rizvi et al., (2017) [23, 24], items were categorized based on their level of discriminatory ability (ranging from items with negative or non-discrimination to excellent discrimination; see Table 3) and a threshold of .30 was used to identify items with good discriminatory ability relative to the item's respective subdomain score or the total score. Specifically, items with a discrimination index of < .30 were flagged for further review and potential removal.

Item standard deviation (SD), reliability index scores, item correlations with total and subdomain scores, and inter-item correlations are provided in supplementary materials. While the current study focused on item reduction and item-level analysis, Cronbach's alpha was also computed for the questionnaire total and subdomains, and is included with the supplementary item-level analysis tables. R 4.0.2 (R Core Team, 2021) [25] and Rstudio 1.4.1106 (RStudio Team, 2021) [26] were used to analyze the data. R's psychometric (v2.2; Fletcher, 2010) package was used to calculate item difficulty index, item discrimination index, and discrimination indices, inter-item correlations, and other item-level metrics [22].

## Results

Patient characteristics are provided in Table 4. The mean age was 60.3 years with SD = 17.3. Summary statistics showed a majority of participants to be male (55.6%), white (66.7%), with nearly all (88%) having at least a high school diploma/GED level education. Most participants reported their annual household income level as either <$20,000 USD (30.8%) or $50,000-$100,00 USD (30.8%). Most participants reported living in a household with at least one other person (78.7%) and the median household size was 2. Participants showed a mean Charlson Comorbidity Index of 2.7 (SD = 2.3) and a mean health literacy of 6.5 (SD = 1.1) on Rapid Estimate of Adult Literacy in Medicine—Short Form (REALM-SF) score.

Examination of full-scale item difficulty and full-scale item discrimination index scores indicated 13 items that could be removed from the full original scale. There was a near complete overlap between items identified by our thresholds based on index scores calculated for the total scale score, and items identified by thresholds for index scores calculated for the three subdomain scores. In other words, most items flagged for removal by the total score item exam were also flagged for removal by subdomain score item exams.

There was also considerable overlap between items identified by item difficulty thresholds and item discrimination thresholds. For the full scale (using total score as criterion) and for the CKD subdomain, only one additional item—which medications can help lower protein in urine—was flagged by the item discrimination index, which the item difficulty did not already identify as potential removal items. For the general kidney knowledge and ESKD knowledge subdomains, all items below discrimination index score threshold had already been identified for removal by their too high or too low scores on item difficulty index.

One item from the ESKD subscale in particular emerged as warranting conditional consideration, however; this item asked about timing to place a fistula. Although this item was just beyond the *a priori* threshold for item difficulty index, it was retained for several reasons. First, although it was outside of the bounds of the item difficulty index (scoring as "too difficult" at .19), it was only just outside the bounds of the threshold by .01. Moreover, it scored as "excellent" (.47) in its ability to discriminate amongst high and low scorers on this disease awareness questionnaire. Finally, there was a substantive rationale to retain this item, as it had a companion question—an item about timing to place a peritoneal dialysis catheter—which met the predefined thresholds for both difficulty and discrimination indices. The content of all other removed items were examined as well, but ultimately their scores on difficulty and discrimination indices resulted in the decision to remove them. Thus, while the team used item difficulty and discrimination index scores as the primary guide for item retention vs. removal, item content and subject matter were also important considerations for each item.

## Discussion

Prior studies have shown that improving patients' knowledge and awareness of their health conditions can have direct positive impacts on their healthcare decisions and clinical outcomes [27]. Similar positive associations have also been shown in patients with CKD. Several studies employing kidney disease education have shown improved CKD awareness positively impacts multiple kidney disease related outcomes, including blood pressure control, vascular access creation, and better control of metabolic imbalances [19, 28–31]. However, these educational domains are focused on individual comorbidities, and substantially different than the knowledge base required to make informed dialysis decision to choose a dialysis modality. Thus, the overall goal of this study is first to create a validated ESKD questionnaire capable of identifying critical deficiencies in individuals' knowledge base essential for forming an informed dialysis decision, and second, to determine whether targeting these domains can lead to greater

**Table 4. Patient characteristics.**

| Characteristic | N = 108[1] |
|---|---|
| Age (years) | 60.3 (17.3) |
| Gender: Female | 48 (44.4%) |
| Race | |
| *Asian* | 3 (2.8%) |
| *Black or African American* | 29 (26.9%) |
| *Hispanic American/Latino* | 4 (3.7%) |
| *White* | 72 (66.7%) |
| Highest Level of Education | |
| *Less than High School Diploma/GED* | 11 (10.5%) |
| *High School Diploma/GED* | 29 (27.6%) |
| *Some College* | 25 (23.8%) |
| *College Degree* | 30 (28.6%) |
| *Advanced Degree* | 10 (9.5%) |
| Household: Pt Lives Alone | 23 (21.7%) |
| Annual Household Income | |
| *<20k* | 32 (30.8%) |
| *20-50k* | 28 (26.9%) |
| *50-100k* | 32 (30.8%) |
| *>100k* | 12 (11.5%) |
| Aware of Kidney Disease | 102 (94.4%) |
| Duration of Awareness of Kidney Disease (months) | 7.7 (8.6) |
| Known Renal Care Duration (months) | 5.3 (5.9) |
| Comfort with Dialysis Decision-Making | |
| *Not comfortable* | 50 (46.7%) |
| *Somewhat Uncomfortable* | 8 (7.5%) |
| *Somewhat Comfortable* | 20 (18.7%) |
| *Very Comfortable* | 29 (27.1%) |
| Current First Choice Dialysis Modality | |
| *Does not want dialysis* | 11 (10.6%) |
| *Home Hemodialysis* | 3 (2.9%) |
| *Home Peritoneal Dialysis* | 37 (35.6%) |
| *In-Center Hemodialysis* | 34 (32.7%) |
| *Needs more information* | 19 (18.3%) |
| Would Consider Transplant | 80 (74.1%) |
| Comorbidities | |
| *Coronary Artery Disease* | 16 (14.8%) |
| *Congestive Heart Disease* | 22 (20.4%) |
| *Peripheral Vascular Disease* | 6 (5.6%) |
| *Cerebrovascular Disease* | 12 (11.1%) |
| *Dementia* | 1 (0.9%) |
| *Chronic Pulmonary Disease* | 12 (11.1%) |
| *Connective Tissue Disease* | 2 (1.9%) |
| *Peptic Ulcer Disease* | 5 (4.6%) |
| *Mild Liver Disease* | 1 (0.9%) |
| *Moderate To Severe Liver Disease* | 4 (3.7%) |
| *Diabetes without End-Organ Damage* | 20 (18.5%) |
| *Diabetes with End-Organ Damage* | 20 (18.5%) |

(*Continued*)

**Table 4.** (Continued)

| Characteristic | N = 108[1] |
|---|---|
| *Hemiplegia* | 0 (0.0%) |
| *Moderate-Severe CKD* | 59 (54.6%) |
| *Local Malignancy* | 14 (13.0%) |
| *Metastatic Malignancy* | 1 (0.9%) |
| *AIDS* | 1 (0.9%) |
| Charlson Comorbidity Index | 2.7 (2.3) |
| Burden of Kidney Disease (KDQoL) | 70.4 (26.8) |
| Effects of Kidney Disease (KDQoL) | 82.5 (20.2) |
| Quality of Life: Physical (KDQoL) | 39.6 (11.7) |
| Quality of Life: Mental (KDQoL) | 49.5 (10.4) |

[1] Mean (SD); n (%)

informed home dialysis utilization. For this initial step of the project, the goal is to evaluate the quality and utility of items for assessing patients' level of kidney disease knowledge.

To ensure that we target knowledge critical for patients to form informed dialysis choice, the study team created a comprehensive questionnaire with inputs from all members of the multidisciplinary CKD team. This resulted in an extensive 45-item questionnaire, whose items were then divided by the authors into 3 domains of general kidney disease awareness, CKD awareness, and ESKD awareness. The questions spanned the entire spectrum of difficulty, with some questions, i.e.,"If you get Medicare and decide to do in-center hemodialysis, when does your Medicare begin?" scoring as difficult as 0.04 on the calculated difficulty (or rather, easiness) index, whereas, others, i.e.,"which one of the following is one of the most common causes of kidney failure worldwide?" scoring as high as 0.88.

Psychometric examination of this scale—in particular, examination of item discrimination and item difficulty—revealed items which could be removed to produce a better instrument. The resultant scale had a total of 32 items, which we believe will be much less burdensome for patients and participants to complete, while still showing a sufficient, non-restricted disease awareness score range and ability to distinguish between respondents with high vs low kidney disease awareness. The resultant scale scores ranged from 0.0% to 87.5% for the Total scale score, while subdomain scores ranged 0.0% to100.0% for the General and CKD subdomains, and 0.0% to 92.3% for the ESKD subdomain (see Table 5). This shows that the scale has a sensitive and large range with neither ceiling nor floor effects.

**Table 5. Patient-level scale & subscale characteristics.**

| | Original Scale | | | | Reduced Scale | | | |
|---|---|---|---|---|---|---|---|---|
| | *n (%)* | *M (SD)* | *Min.* | *Max.* | *n (%)* | *M (SD)* | *Min.* | *Max.* |
| Total Score | 108 | 19.7 (9.2) | 0 | 36 | 108 | 15.2 (7.8) | 0 | 28 |
| Percent Correct | | 43.7% (20.4%) | 0.0% | 80.0% | 108 | 47.5% (24.4%) | 0.0% | 87.5% |
| Subdomain Scores | | | | | | | | |
| General | 108 | 6.8 (2.5) | 0 | 10 | 108 | 3.2 (1.5) | 0 | 5 |
| Percent Correct | | 67.5% (25.2%) | 0.0% | 100.0% | | 63.9% (29.0%) | 0.0% | 100.0% |
| CKD | 108 | 8.0 (4.4) | 0 | 20 | 108 | 7.1 (3.9) | 0 | 14 |
| Percent Correct | | 37.9% (21.0%) | 0.0% | 95.2% | | 50.7% (27.7%) | 0.0% | 100.0% |
| ESKD | 108 | 4.9 (3.6) | 0 | 12 | 108 | 4.9 (3.5) | 0 | 12 |
| Percent Correct | | 35.3% (25.6%) | 0.0% | 85.7% | | 37.7% (27.3%) | 0.0% | 92.3% |

While the items in this instrument were written within a structure of 3 subdomains, including general kidney knowledge and CKD knowledge, the ESKD subdomain was of particular interest, as noted above. Thus, the relatively high proportions of ESKD items (number of ESKD items retained relative to number of General and CKD items retained, and relative to the total number of items in the questionnaire) is ideal for our purposes. The ESKD subdomain had only two items identified for removal based on discrimination and difficulty index scores, but one was retained (see explanation for fistula item above), resulting in 13/14 = 92.9% of the ESKD items retained.

These items together constitute an instrument that shows strong ability to distinguish between high and low disease awareness. There is yet much to learn, however, about the impact of kidney disease awareness on patient outcomes. First, it is still unknown whether this instrument would show similarly high ability to discriminate between high and low disease awareness in other samples of kidney disease patients. Furthermore, it is unclear whether disease awareness is related to ability to make informed decisions about disease management—for example, whether it is predictive of patients' disease management confidence, attitudes, and behaviors. It would be especially useful to know whether kidney disease awareness, especially ESKD awareness, was predictive of dialysis modality (e.g., in-center vs home dialysis) selection and long-term use.

Assuming that a patients' disease awareness is related to disease management down the line, it is also unknown whether disease awareness can be improved. Providing comprehensive kidney disease education could increase disease awareness, which should be related to greater ability for patients to make the best choices for themselves and their individual lifestyles regarding disease management. This increased capacity for informed decision making could in turn translate to improved disease management choices and behavior, as alluded to previously, which may have potentials to ultimately influence the quality of life. Thus, development of a questionnaire to measure patients' kidney disease awareness is an early and important step for assessing the effects of disease awareness, and potentially improving not only patients' disease awareness but disease outcomes.

There are several limitations to the current work on developing an ESKD-focused kidney disease awareness questionnaire. Boateng et al. (2018) outlined nine steps for scale validation and development, and not all steps have been completed here [32]. For example, while items were developed with consideration from multiple experts in specialty renal care, cognitive interviews were not conducted with a sample from the target audience (kidney disease patients). We plan to address as many of these shortcomings as possible, however, in subsequent work. With the items developed for this questionnaire, future directions for this multipart study including further testing and validation of the instrument itself (e.g., factor analysis of the proposed 3 domain structure), as well as testing against other patient and clinical factors. Specifically, we will test the validity and predictive ability of this questionnaire against quality-of-life measures and other hard outcomes (e.g., dialysis choice and dialysis use). We will test whether kidney disease awareness is associated with patient characteristics such as age, education, or comorbidity. Finally, we will test whether kidney disease awareness can be improved via patient education.

Second, the survey items were developed during the period when there was some ongoing debate in the kidney disease community about patient-centered verbiage—for example, the use of "kidney" instead of "renal" or "nephro-." As such, items were developed during this time to assess patient comfort with such verbiage. Some such items with older terminology were retained in the reduced questionnaire, but for future implementation of the survey, we will replace "renal" and "nephro-"with "kidney" where possible and appropriate.

Finally, while a 32-item questionnaire would be less burdensome for patients than a 45-item questionnaire, the length of this measure may yet be a barrier to its widespread use. Many routinely used instruments in the kidney disease community, i.e., KD-QOL (36 items) and KiKS survey (28 items) have significant item length as well. We anticipate that our future work (such as factor analysis) may result in further refinement and reduction of items, or result in the construction of short forms for use when the patient has other questionnaires to complete, reserving the full scale for research or clinical situations which require fewer questionnaires. Moreover, further validation of the questionnaire may support administration of items belonging to a specific domain of interest for the practitioner. For example, the clinician may wish to focus on, and thus only administer, the ESKD domain of the questionnaire, which at 13-items may improve its feasibility in routine clinical use. In the meantime, it may be useful to nevertheless provide the current questionnaire for practitioners and researchers who wish to assess and improve patients' kidney disease awareness.

To summarize, previous research has shown that improving patients' disease awareness can have positive impacts on healthcare decisions and clinical outcomes. Effectively studying this in kidney disease, and more specifically in ESKD requires an instrument that can accurately assess ESKD specific knowledge. This study seeks to address this existing gap. Developing a targeted questionnaire to assess ESKD knowledge can accurately define the deficiencies in patients' ESKD awareness. It can further allow assessment of the efficacy of measures aimed to improve these deficiencies and facilitate better targeting of patient education efforts in this area to address the long-term nephrology goals for increased informed home dialysis use.

## Supporting information

**S1 Appendix. Inter-item correlations original.**
(XLSX)

**S1 File. Item level analyses.**
(DOCX)

**S2 File. ESKD awareness questionnaire.original.FULL.**
(DOCX)

**S3 File. ESKD awareness questionnaire.original.Reduced.**
(DOCX)

## Author Contributions

**Conceptualization:** Tatiana Orozco, Emma Segal, Colin Hinkamp, Olanrewaju Olaoye, Popy Shell, Ashutosh M. Shukla.

**Data curation:** Tatiana Orozco, Emma Segal, Colin Hinkamp.

**Formal analysis:** Tatiana Orozco.

**Funding acquisition:** Ashutosh M. Shukla.

**Investigation:** Tatiana Orozco, Emma Segal, Colin Hinkamp, Ashutosh M. Shukla.

**Methodology:** Tatiana Orozco, Colin Hinkamp, Olanrewaju Olaoye, Ashutosh M. Shukla.

**Project administration:** Tatiana Orozco, Ashutosh M. Shukla.

**Resources:** Olanrewaju Olaoye.

**Software:** Tatiana Orozco.

**Supervision:** Olanrewaju Olaoye, Ashutosh M. Shukla.

**Validation:** Tatiana Orozco, Ashutosh M. Shukla.

**Visualization:** Ashutosh M. Shukla.

**Writing – original draft:** Tatiana Orozco, Olanrewaju Olaoye, Popy Shell, Ashutosh M. Shukla.

**Writing – review & editing:** Tatiana Orozco, Olanrewaju Olaoye, Popy Shell, Ashutosh M. Shukla.

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
