## [Decision Letter · Decision Letter 0]

11 Jan 2022

PONE-D-21-34931Development and Validation of an End Stage Kidney Disease Awareness Survey: Item Difficulty and Discrimination IndicesPLOS ONE

Dear Dr. Shukla,

Thank you for submitting your manuscript to PLOS ONE. After careful consideration, we feel that it has merit but does not fully meet PLOS ONE’s publication criteria as it currently stands. Therefore, we invite you to submit a revised version of the manuscript that addresses the points raised during the review process.

We look forward to receiving your revised manuscript.

Kind regards,

Abduzhappar Gaipov

Academic Editor

PLOS ONE

Journal Requirements:

“Shukla AM reports funding from Department of Veterans Affairs, Health Service Research and Development (I01HX002639), Office of Rural Health, and Clinical Science Research and Development (I01CX001661).”

4. We note you have included a table to which you do not refer in the text of your manuscript. Please ensure that you refer to Table 3 in your text; if accepted, production will need this reference to link the reader to the Table.

Reviewers' comments:

Reviewer's Responses to Questions

**Comments to the Author**

1. Is the manuscript technically sound, and do the data support the conclusions?

Reviewer #1: Yes

Reviewer #2: Yes

2. Has the statistical analysis been performed appropriately and rigorously? 

Reviewer #1: Yes

Reviewer #2: Yes

3. Have the authors made all data underlying the findings in their manuscript fully available?

Reviewer #1: Yes

Reviewer #2: Yes

4. Is the manuscript presented in an intelligible fashion and written in standard English?

Reviewer #1: Yes

Reviewer #2: Yes

5. Review Comments to the Author

Reviewer #1: The authors did a great work creating such an excellent questionnaire to assess ESKD awareness among kidney patients. There are, however, some points to be revised.

1. The full scale questionnaire included questions that are currently under a strong debate among the nephrology community. The use of word renal and nephrology are starting to vanish. Q2 and Q15 asked about these items in particular. While there is an international movement towards the use of kidney instead of renal to be more close to the patient understanding. For example we moved from acute renal failure to acute kidney injury, and from end-stage renal disease (ESRD) to end-stage kidney disease (ESKD) long time ago, however, authors are still using these old terminology. In the latest publication of KDIGO, they recommended clearly to use "kidney" rather than "renal" or "nephro-"

https://pubmed.ncbi.nlm.nih.gov/32409237/

Is it appropriate to ask patients about term that will be recently changed to make it more clear for patients' understanding.

2. Checking the reliability score from S2 item level analysis provided by the authors ... I find the reliability score are far below accepted standards worldwide. All scores are below 0.5 in the full scale, reduced scale total scores and subdomain scores!

The authors didn't discuss these results in their manuscript or explain these low reliability score levels ?!

3. Please provide the reduced scale questionnaire including the 32 items. The reader will be interested in knowing which items were removed and what are the remaining items ?

4. The authors stated in their limitations sections that there are many shortcoming and that they are planning to address

as many of these shortcomings as possible, however, in subsequent work. I understand and accept that the test be tested for correlation with hard outcomes in further studies. However, if there are additional steps for scale validation needed, why weren't they finished before publishing this manuscript? The authors satated this clearly in their limitations section: "Boateng et al. (2018) outlined nine steps for scale validation and development, and not all steps have been completed here."

Reviewer #2: In the current manuscript, Orozco et al developed a questionnaire representing “patient disease awareness instrument”, as the first step of a multistep project. Initially, the questionnaire included 45 items distributed under 3 subdomains. Considering item difficulty and item discrimination index scores, it was then reduced to 32 items, (5/10 items in the general kidney knowledge subdomain, 14/21 items in the CKD knowledge subdomain, and 13/14 items in the ESKD knowledge subdomain).

Below are my comments:

1- The manuscript focused only on the development of the survey as an initial step of a multistep project. However, authors stated the purpose of the project as a whole and did not specifically address the aim of the manuscript, which can be confusing for the reader (page /9/23; last paragraph).

2- The item difficulty index, if I understand well, was calculated as c/n, where c the number of correct answers to a question and n the total number of respondents to that question i.e., the percent of correct answers (page 11/23).

3- Appendix 1 included all the 45 items. I suggest identifying the removed items and distributing the questions under the 3 subdomains.

4- I suggest it would be more interesting to define participant demographics and characteristics.

5- The practicality of using a relatively long survey of 32 MCQ items in every day clinical practice need to be discussed.

6. PLOS authors have the option to publish the peer review history of their article (what does this mean?). If published, this will include your full peer review and any attached files.

Reviewer #1: **Yes: **Mohamed E Elrggal

Reviewer #2: No

---

## [Author Response · Author response to Decision Letter 0]

22 Mar 2022

The response to reviewer document has been uploaded along with the manuscript.

---

## [Decision Letter · Decision Letter 1]

23 May 2022

Development and Validation of an End Stage Kidney Disease Awareness Survey: Item Difficulty and Discrimination Indices

PONE-D-21-34931R1

Dear Dr. Shukla,

We’re pleased to inform you that your manuscript has been judged scientifically suitable for publication and will be formally accepted for publication once it meets all outstanding technical requirements.

Kind regards,

Abduzhappar Gaipov

Academic Editor

PLOS ONE

Additional Editor Comments (optional):

Reviewers' comments:

Reviewer's Responses to Questions

**Comments to the Author**

1. If the authors have adequately addressed your comments raised in a previous round of review and you feel that this manuscript is now acceptable for publication, you may indicate that here to bypass the “Comments to the Author” section, enter your conflict of interest statement in the “Confidential to Editor” section, and submit your "Accept" recommendation.

Reviewer #1: All comments have been addressed

Reviewer #2: All comments have been addressed

2. Is the manuscript technically sound, and do the data support the conclusions?

Reviewer #1: Yes

Reviewer #2: Yes

3. Has the statistical analysis been performed appropriately and rigorously? 

Reviewer #1: N/A

Reviewer #2: Yes

4. Have the authors made all data underlying the findings in their manuscript fully available?

Reviewer #1: Yes

Reviewer #2: Yes

5. Is the manuscript presented in an intelligible fashion and written in standard English?

Reviewer #1: Yes

Reviewer #2: Yes

6. Review Comments to the Author

Reviewer #1: (No Response)

Reviewer #2: The authors did great work and developed ESKD awareness questionnaire as the first step of a multistep project with the ultimate goal of improving healthcare decisions and clinical outcomes in patients with ESKD. I'd like to thank them for the comprehensive and clear discussion of all points raised in the review.

7. PLOS authors have the option to publish the peer review history of their article (what does this mean?). If published, this will include your full peer review and any attached files.

Reviewer #1: **Yes: **Mohamed Elrggal

Reviewer #2: **Yes: **Ali M Shendi

---

## [Editor Report · Acceptance letter]

28 Jul 2022

PONE-D-21-34931R1 

Development and Validation of an End Stage Kidney Disease Awareness Survey: Item Difficulty and Discrimination Indices 

Dear Dr. Shukla:

I'm pleased to inform you that your manuscript has been deemed suitable for publication in PLOS ONE. Congratulations! Your manuscript is now with our production department. 

Kind regards, 

on behalf of

Dr. Abduzhappar Gaipov 

Academic Editor

PLOS ONE